# Analysis of the Technical Condition of a Late 19th Century Public Building in Łódź

**DOI:** 10.3390/ma16051983

**Published:** 2023-02-28

**Authors:** Wioletta Grzmil, Justyna Zapała-Sławeta, Jagoda Juruś

**Affiliations:** Faculty of Civil Engineering and Architecture, Kielce University of Technology, 25-314 Kielce, Poland

**Keywords:** heritage structures, concrete, strength, carbonation, water absorption

## Abstract

Heritage building structures in many situations contribute unique value to national cultural heritage. In engineering practice, monitoring of historic structures includes visual assessment. This article assesses the condition of the concrete in one of the most recognizable buildings in Łódź, the former German Reformed Gymnasium, located at Tadeusza Kościuszki Avenue. The paper reports a visual assessment of the structure and the degree of technical wear affecting selected structural components of the building. A historical analysis of the building’s state of preservation, characterization of the structural system, and an assessment of the condition of the floor-slab concrete were carried out. The state of preservation of the eastern and southern facades of the building was found to be satisfactory, while on the western side (with the courtyard) the facade is in a poor state of preservation. Tests were also conducted out on concrete samples taken from individual ceilings. The concrete cores were tested for compressive strength, water absorption, density, porosity, and carbonation depth. Corrosion processes including the degree of carbonization and the phase composition of the concrete were identified by X-ray diffraction. The results are indicative of the high quality of the concrete produced more than 100 years ago.

## 1. Introduction

Historic building structures are often of unique value to national cultural heritage. When a building is in use, it is sometimes periodically out of service. The reasons for this can vary and may be affected by the socioeconomic conditions in which the building is used. Monitoring of a historic building structure can take the form of periodical or comprehensive inspection of the technical conditions of the structure, for example, in the context of a planned upgrade. In engineering practice, the monitoring of historic structures includes a visual assessment. Visual assessment is usually the first step in the diagnosis of any structure [1]. A general inspection of a building structure can require a very different approach depending on whether it is possible to conduct non-destructive or semi-destructive testing of the structure [2,3,4,5,6,7]. In some cases, tests conducted directly on the structure, i.e., in situ, require supplementation with laboratory tests, such as strength and material tests. 

The analysis of the condition of a public utility building was carried out for one of the most recognizable buildings in Łódź, the former German Reformed Gymnasium located at Tadeusza Kościuszki Avenue. Due to its historical and historic nature, prior to the examination of the structural elements, it was necessary to carry out historical analysis involving visual assessment of the building, the condition of the façade, and the nature of its structure.

The work included a visual assessment of the object and the degree of technical wear affecting a selected structural element. Historical analysis was conducted to assess the state of preservation of the building and the characteristics of the structural system. Structural diagnosis is an integral part of the above-mentioned tests.

## 2. Scope and Purpose of Research 

Analysis of the technical condition of the facility was carried out due to its historic character. As the literature indicates, the diagnostics of building objects is a complex task due to the unique nature of the objects, and in many cases requires an individual research approach [8,9,10]. It should be noted that in the analysis of the condition of historical objects, research techniques and methods that do not threaten the structural integrity are recommended [11,12,13,14].

When planning the analysis of the technical condition of the building, access to individual structural elements was taken into account, which can often be limited due to geometry and location. Currently, preparatory works are being carried out on the premises, as part of the planned reconstruction and modernization of the facility. The purpose of this research was to analyze the technical conditions, and the study process was divided into three main areas: historical analysis, analysis of the existing conditions, and assessment of concrete properties, according to the scheme shown in Figure 1.

## 3. Historical Analysis, Characteristics of the Structural Layout, and Technical Condition of the Building

### 3.1. Historical Analysis

A former school building, the German Reformed Gymnasium (Figure 2) is located in the downtown area of Łódź, at 65 Tadeusza Kościuszko Avenue. It exemplifies features of Berlin Modernism and elements of Art Nouveau (Figure 3). The building isincluded in the register of historical buildings of the city of Łódź, item number A/307.

The main school building (building A, Figure 4) was constructed between 1909 and 1910. The building was originally designed on an inverted L-letter plan and its outline varied with avant-corps of different depths. The design was the work of the well-known Berlin architect Otto Herrnring, who specialized in the design of educational buildings; as a result, the building has been called a “model” school building of the time. The structure was one of the first public buildings in Łódź to use a reinforced concrete monolithic frame.

Over the subsequent years, the building changed ownership and was expanded and upgraded. The structure was expanded on the north side in the late 1930s, with the addition of a three-floor building (building B, Figure 4) that filled the gap between the existing buildings and formed a continuous alignment. The German Gymnasium existed until 1945, then after World War II the building housed a high school. Between 1945 and 1972, the structure was expanded with the addition of two more buildings (buildings C and D, Figure 4). From 1973 to 2015, the University of Łódź owned and used the building. During that period, the structure underwent considerable renovation work including the adaptation of individual rooms. In 2015, the structure became the property of the Court of Appeals in Łódź.

### 3.2. Characteristics of the Structural System

A four-floor building in its southern part, with a usable attic and basement, and a three-floor building in the northern avant-corps with a usable attic and no basement. To the north, a four-floor addition, with no basement or attic, connecting the courthouse to the adjacent bank building. On the southeast corner, the dominant feature is an an astronomical observatory tower.

The building is a reinforced concrete monolithic frame structure with fields filled with solid red ceramic brick masonry, bonded with lime mortar, with masonry load-bearing walls, masonry chimneys, and concrete strip foundations.

Above the basement, the ground floor, the second and third floors, and pathways other than corridors, there is reinforced concrete slab-and-rib ceiling, constructed of a 15 cm thick slab supported on load-bearing walls, with 25 × 30 cm ribs supported in the axis of the inter-window pillars of the exterior walls at axial spacing of 2 m to 2.40 m. Above the circulation zones there are 15 cm thick reinforced concrete slab ceilings supported on the longitudinal walls. Above the third floor, there is a wooden ceiling made of 26 cm high beams up to 100 cm apart and supported on the longitudinal load-bearing walls. Klein ceilings are present in the connecting structure between the main building and the bank building. Reinforced concrete footings are covered with cement-lime mortar.

Other features include a mansard roof, covered with sheet metal, and wooden, purlin, and rafter roof trusses supported on the final ceiling and exterior walls, originally covered with red clay tiles, now with flat zinc sheets with standing seam joints on openwork boarding in a French pattern (rhomboids) between multi-sash windows [17].

### 3.3. An Analysis of the Technical Condition of the Building

The former school building is one of the most recognizable buildings in Łódź, its architecture corresponding to the trends of Berlin architecture at the time of its construction. It is a corner building with the eastern front facade facing Kościuszki Avenue and the southern facade facing Zamenhofa Street.

The object is preserved in its original state. It survived the First and Second World Wars virtually intact, and the structure of the object is currently intact. The source materials mention only replacement of the roofing (in the years 1945-72), i.e., the removal of ceramic tiles, probably destroyed during the war, with zinc sheeting. Due to its historical nature, it was necessary to carry out a visual assessment of the facility, the condition of the facade and the structural system. 

An analysis of the condition of the building facade, located in the city centre directly among streets with heavy traffic, showed that the building has been exposed to pollutants from the air, i.e., car-exhaust fumes, and in previous years also industrial fumes containing sulfur and nitrogen oxides, as well as dust and soot from buildings using coal-fired stoves. This has resulted in dark or black deposit layers on the facades. The deposits have had a destructive effect on porous materials, e.g., bricks, plaster, and roof tiles (Figure 5).

The facades show numerous cracks in the masonry and cavities. The plaster is in a very bad condition. Superficial defects in the plaster are visible, due to the large cavities and damaged flashings (i.e., window sills and gutters), which have caused individual layers to separate, exposing the bricks in permanently damp areas. The increased moisture from rainwater has caused biological corrosion of the plaster, manifesting itself in crumbling, delamination, and peeling, and consequently in the plaster falling off the brick wall (Figure 6).

Corrosion of the reinforced concrete structure is visible on the eastern and southern facades. The layer around the reinforcement has been destroyed there, and the corroded rebar is visible (Figure 7).

Numerous cavities and areas replastered with secondary cement mortar are visible on all facades of the building (Figure 8).

The windows, a large percentage of which are the original ones, present a varying degree of preservation. Due to the effects of the weather conditions, the paint coatings have become cracked and chipped, and have been subject to numerous secondary coatings. There is visible unsealing of the joinery joints, damage to the wood and the paint coatings, as well as numerous paint coats that are clearly detached from the wooden structure of the windows.

The exterior stairs, on the eastern and southern facades, were constructed of red sandstone. Their surface shows damage and cavities, and there are mosses, algae, and lichens due to the long-term presence of moisture (Figure 9).

The architectural details on the building’s facades are very rich and varied. In terms of their construction material, they can be divided into two groups. The first group includes moldings made of cement–lime mortar, and the second group consists of reliefs made of artificial stone (gypsum mortar). 

The architectural details are in a good state of preservation. The top surfaces of the cement–lime mortar reliefs show superficial cavities in the plaster, damp spots, and cracks.

The state of preservation of the eastern and southern facades of the building is satisfactory, while on the western side (the side of the courtyard) the facade presents a poor state of preservation.

## 4. The Scope and Results of the Tests Performed on the Concrete from Floor Slabs

An external visual inspection of the condition of the structure indicated damage to the building’s facade. The cause of the damage shown in the example Figure 4 and Figure 5 may be the lack of weatherproofing and the resulting dampness of the building components.

An important part of the material diagnostics is an assessment of their condition within the structure. Progression of corrosion processes depends primarily on the internal structure of the medium, namely the hardened structure of concrete, and the amount of moisture it contains [18]. Corrosion processes can cause adverse changes in the structure of concrete by increasing its porosity. Under the influence of, for example, mechanical dead loads, micro-cracks form that facilitate the migration of corrosive agents into the concrete. Inside buildings, concrete is particularly exposed to CO_2_. An important part of concrete diagnostics, especially in a more than 100-year-old structure, is the evaluation of its physical–mechanical and durability characteristics, including tests of the progress of carbonation in floor slabs. 

Since the building required upgrading after such a long period of use, a study was conducted to determine the condition of the structural components. This paper presents the test results forthe floor slab components. 

### A Description of the Tests and the Methods

A compressive strength test and a saturation test were carried out on samples taken using a core drill. Six core borings were taken from each floor slab on every floor. Since the building is supervised by a conservation officer, the opportunity to take samples was limited to only the selected sites that were to be demolished. The locations where the concrete core samples were taken are shown in Figure 10.

The tests were conducted on core samples of diameter 100 mm and length 100 mm. The number of borehole samples that may be taken from historic structures may be limited; however, according to [19,20], it can range from 3 to 14 samples. The strength test was performed on the borehole samples according to standard [21]. The samples were prepared according to the requirements set forth in the standard, by grinding the outer surfaces of the cores to ensure that they were parallel. Figure 11 shows a core borehole in a strength-testing machine. 

The weight and volumetric absorption and the bulk density of the concrete were determined for the core samples, according to standards [22]. 

Weight water absorption as the ratio of the mass of water absorbed to the mass of the dry material sample was determined using Formula (1):(1)$$w_{A}=\frac{w_{w}-w_{d}}{w_{d}}100\%$$

$w_{A}$—weight water absorption,
$w_{w}$—weight of the sample when saturated with water
$w_{d}$—dry mass of the sample

Volumetric water absorption as the ratio of the volume of water contained in the tested sample of material to the volume of this sample in a dry state, was calculated according to (2):(2)$$w_{V}=\frac{w_{w}-w_{d}}{V}100\%$$

$w_{v}$—volumetric water absorption
$w_{w}$—mass of the sample saturated with water
$w_{d}$—weight of the sample dried to constant weightV—volume of the sample

The open porosity test was performed in accordance with the standard [23]. The test consisted in saturating the samples in a vacuum. After about 30 min of vacuum under negative pressure, the samples were soaked in degassed distilled water heated to about 80 °C. The open porosity was determined using the Formula (3):(3)$$p_{o}=\frac{w_{wp}-w_{d}}{V}100\%$$

$p_{o}$—open porosity
$w_{wp}$—mass of sample saturated with hot water after vacuuming,$w_{d}$—weight of the sample dried to constant weight,V—volume of the sample

The depth of carbonation was measured on fresh fractures of core borehole samples by spraying a phenolphthalein solution, according to standard [24]. 

The samples for phase composition analysis were crushed and the coarse aggregate was separated. The mineral composition of the aggregate itself was determined. Materials for analysis in the form of powder were ground manually in an agate mortar to a size less than 63 μm. Tests were conducted in an Empyrean X-ray diffractometer (from PANalytical, Almelo, The Netherlands) using a Cu lamp and an X’Celerator detector in the 2θ angle range from 5 to 75°2θ. The ICDD PDF-2 database was used as a reference database for the diffractogram analysis [25]. 

## 5. Test Results

### 5.1. Tests of Physical and Mechanical Characteristics of Concrete

The average results of the determination of physical and mechanical characteristics performed on the samples are shown in Table 1 and Figure 12.

The concrete samples taken from individual floor slabs had very similar physical and mechanical characteristics. The weight absorption was in the range of 3.03 to 3.30% and the volumetric absorption was in the range of 7.43 to 8.00%. The concrete from floor slab 2 had the lowest weight absorption. The literature recommends limiting the weight absorption of concrete to a certain level w_A_ ≤ 4% [26,27] 

The concrete in the analyzed floors showed a similar bulk density of about 2500 kg/m^3^. The open porosity of the concrete samples varied, ranging from 10.70 to 12.04%. The air content of the different concrete samples, as determined by measuring the open porosity, was similar in the range 3.26 to 4.03%. The results obtained may indicate different degrees of porosity in the analyzed concrete samples.

The average compressive strength (for three samples) determined for the three analyzed floors was in the range of 27.5 to 36.5 MPa. 

The relationships between the weight absorption and the compressive strength were determined. The results of the analyses are shown in Figure 13.

The weight absorption of the concrete samples from floor slabs 1 and 2 was similar, and ranged from 2.8 to 3.2%, but the samples had different compressive strengths. The concrete samples from floor 3 had the highest compressive strength and weight absorption among all the samples.

### 5.2. Assessment of the Corrosion Hazard

To determine the corrosion hazard to the reinforcing steel, the degree of carbonation of the concrete was measured and the phase composition was determined by X-ray diffraction.

The tests showed that the thickness of the near-surface layer of concrete that was not stained in the phenolphthalein test reached 3 mm in the top part of the floor slab, while the depth of concrete carbonation in the bottom part of the floor slab was 10 mm (Figure 14). The top layer of the floor slab had been covered with xylolith flooring that contained magnesia cement, which may have influenced the small depth of carbonation.

The analyzed concrete consists of a cement binder and natural aggregate, showing variations under macroscopic evaluation. An X-ray analysis indicated that the concrete was composed of aggregate from sedimentary rocks, mainly limestone (calcite), feldspar (anorthoclase), and quartz. Clay minerals were identified, i.e., illite, found in small amounts, as well as muscovite, a common rock-forming mineral (Figure 15).

Analysis of the phase composition of the cement slurry demonstrated the presence of polymorphic varieties of calcium carbonate, i.e., calcite, vaterite, and aragonite (Figure 16 and Figure 17).

Vaterite is reported to be a product of the reaction of CO_2_ with portlandite, both of which are formed in carbonated concrete, and transforms into calcite [28]. Despite the presence of polymorphic forms of calcium carbonate, reflections of 18° 2θ and 34° 2θ were detected, indicating the presence of uncarbonated portlandite.

Differences in the intensity of the reflections of portlandite, vaterite, and calcite were observed. The intensity of the portlandite reflections was lower in the concrete taken from the bottom layer of the floor slab, with simultaneously less intense vaterite reflections and higher intensity of calcite reflections comoared with the concrete from the top layer of the floor slab. The results indicate a higher degree of concrete carbonation processes in the bottom layer of the floor slab.

## 6. Summary and Conclusions

A former school building, the German Reformed Gymnasium, located in the downtown area of Łódź, at 65 Tadeusza Kościuszko Avenue, is an example of a building with features of Berlin Modernism and elements of Art Nouveau. The state of preservation of the southern and eastern facades of the building is satisfactory, while on the western side (with the courtyard side) the facade presents a poor state of preservation. 

An important part of the diagnostics of reinforced concrete structures of historical value is an assessment of the corrosion hazard to reinforcing steel. The tests and measurements that were carried out indicate that the floor slabs are in a good condition despite their long service life. From the point of view of the durability of the reinforced concrete structure, the process of carbonation is not advanced, as evidenced by the comparatively thin near-surface layer of concrete that was unstained in the phenolphthalein test. It should be noted that the thickness of the carbonated layer depends on time. According to the Verbeck relationship cited by Kurdowski, the thickness of the layer after 100 years can be about 5 cm [28].

The slow progression of carbonation processes in the floor slab concrete is also indicated by the phase composition of the cement slurry, in which portlandite was detected by X-ray radiography despite the concrete’s age of 100 years. In addition, low values of absorbability by weight (in the range of 3.03 to 3.26), under 5% in all the floor slabs tested, testify to the tightness and high homogeneity of the concrete structure.

Nowadays, when designing concretes, the type of structural component and the classes of exposure affecting the concrete are taken into account. Analysis of the strength properties of the tested concrete and the contemporary requirements [29] indicates that the minimum class of concrete for a floor slab should be C20/25. The estimated class of the concrete from the late 19th century floor slabs, as determined for the core samples, is C25/30 for floor slabs 1 and 2, and C 30/37 for floor slab 3 [30]. The specified strength classes meet the requirements for floor slabs in accordance with contemporary design recommendations.

Based on the test results obtained, it can be concluded that this concrete produced more than 100 years ago is of high quality.

## Figures and Tables

**Figure 1 materials-16-01983-f001:**
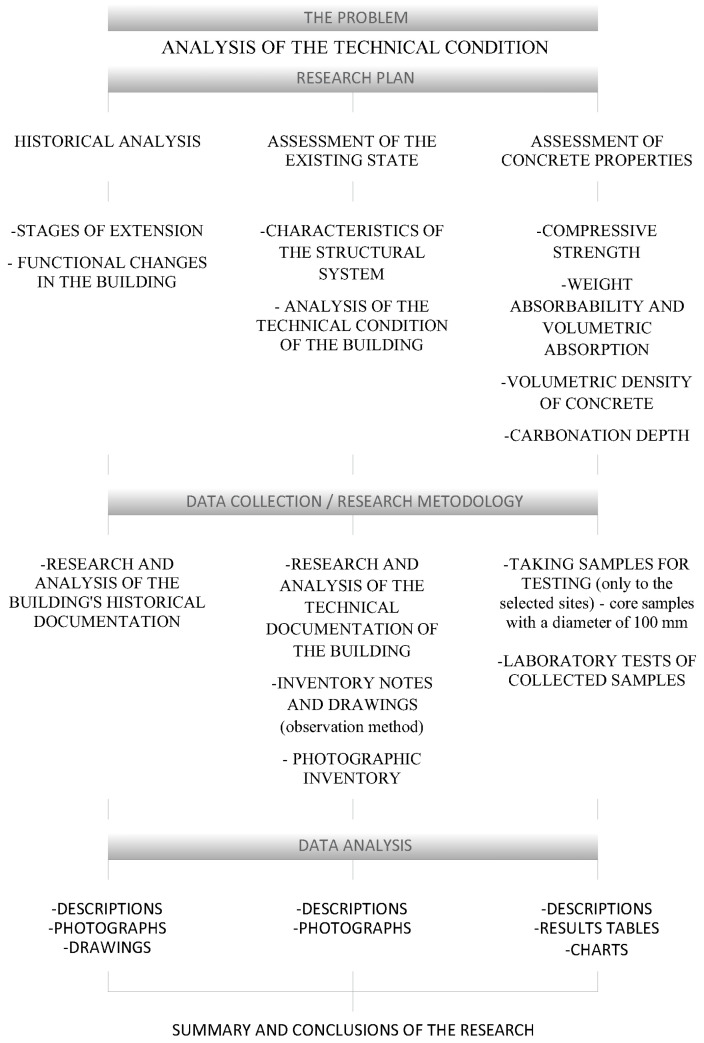
Diagram of the research process.

**Figure 2 materials-16-01983-f002:**
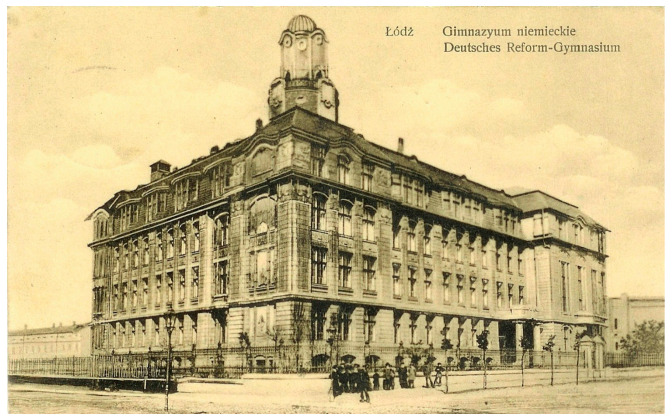
The German Gymnasium, 1908 [15].

**Figure 3 materials-16-01983-f003:**
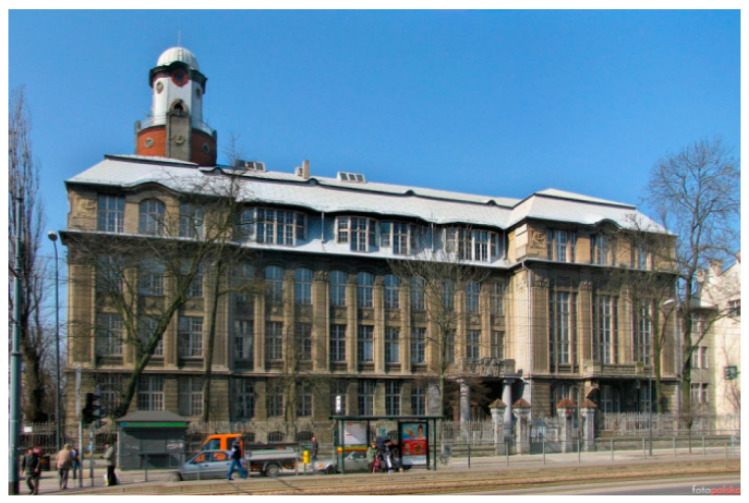
The view of the eastern facade of the building, then the Faculty of Philosophy of the University of Łódź; 2015 [16].

**Figure 4 materials-16-01983-f004:**
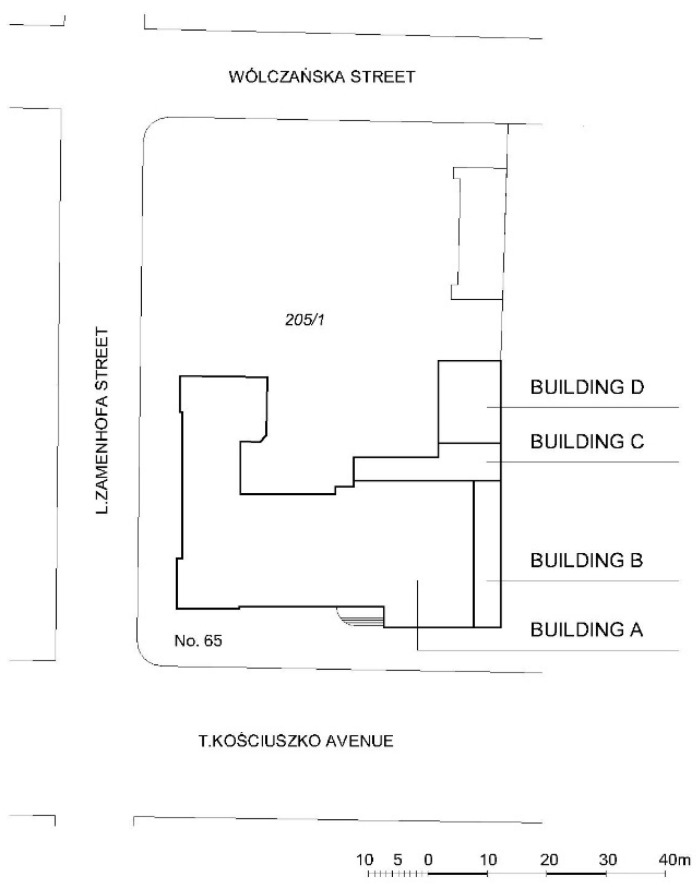
Site: the current layout of the building.

**Figure 5 materials-16-01983-f005:**
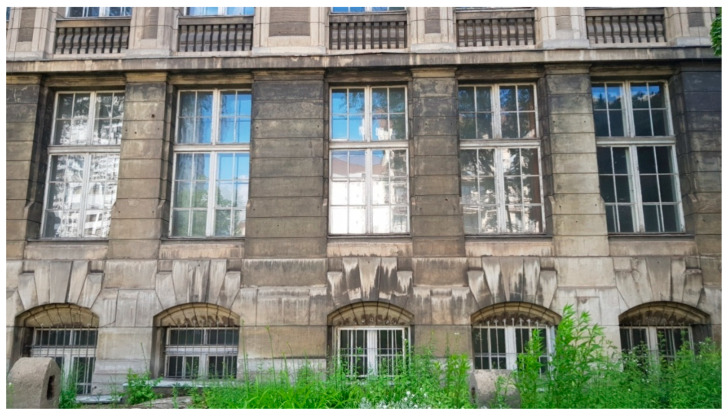
Part of the southern façade, with dark and black deposits visible.

**Figure 6 materials-16-01983-f006:**
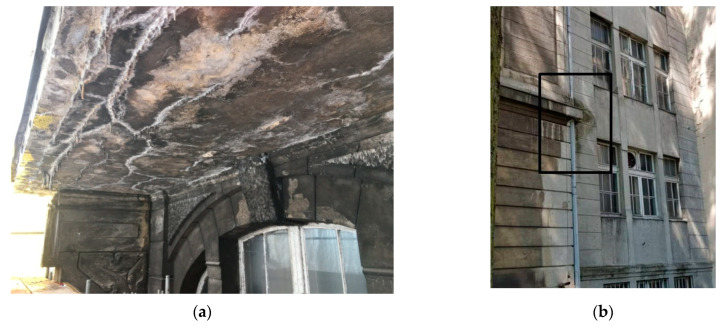
Dampness of the balcony (**a**) and the building walls (**b**).

**Figure 7 materials-16-01983-f007:**
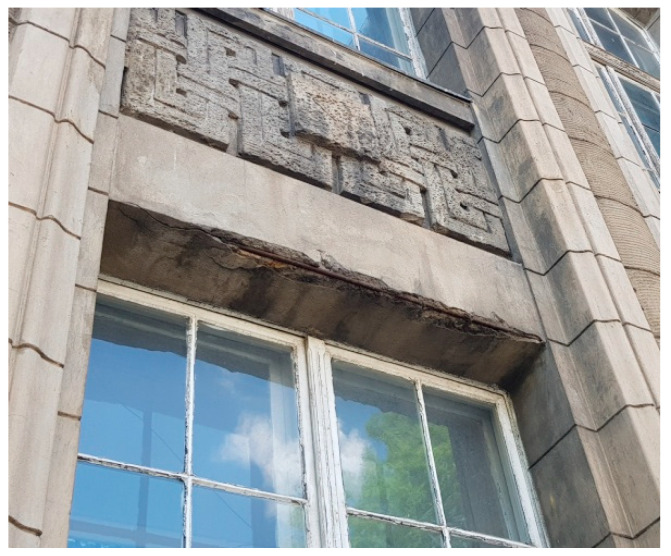
Damage to the layer around the reinforcement—visible rebars.

**Figure 8 materials-16-01983-f008:**
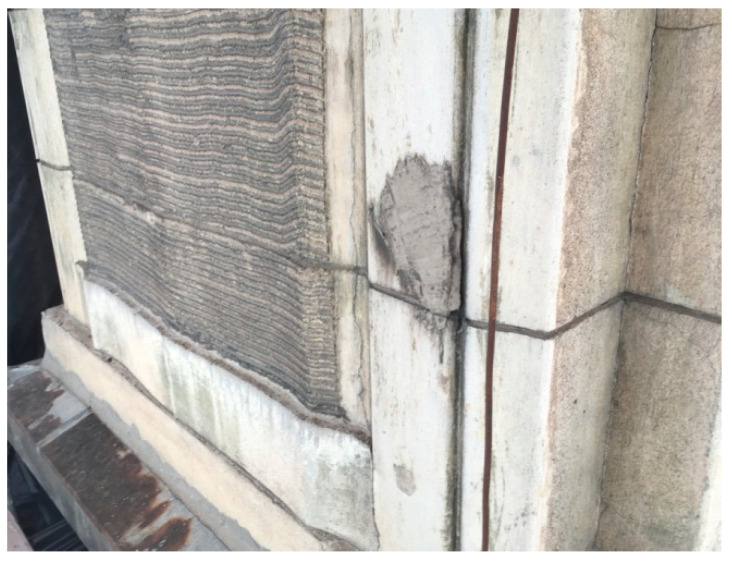
Cavities and areas replastered with secondary cement mortar.

**Figure 9 materials-16-01983-f009:**
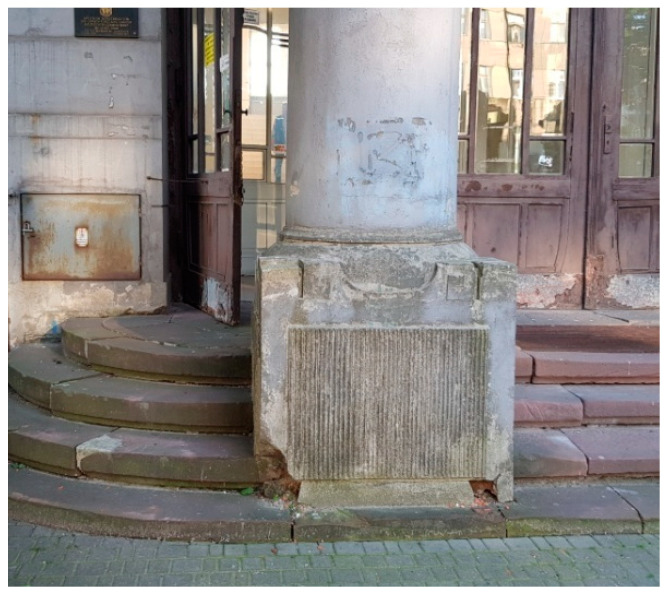
Damage and cavities in exterior stairs made of red sandstone.

**Figure 10 materials-16-01983-f010:**
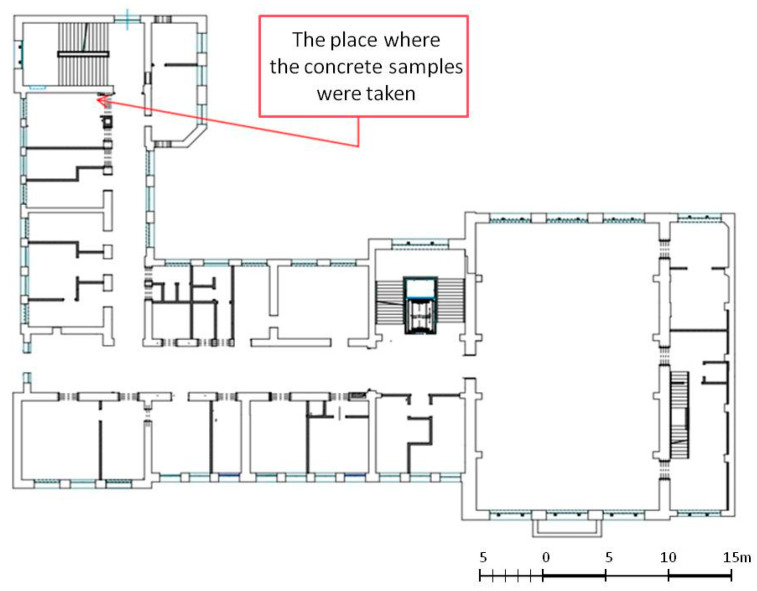
The locations in the building from which the concrete samples were taken for testing.

**Figure 11 materials-16-01983-f011:**
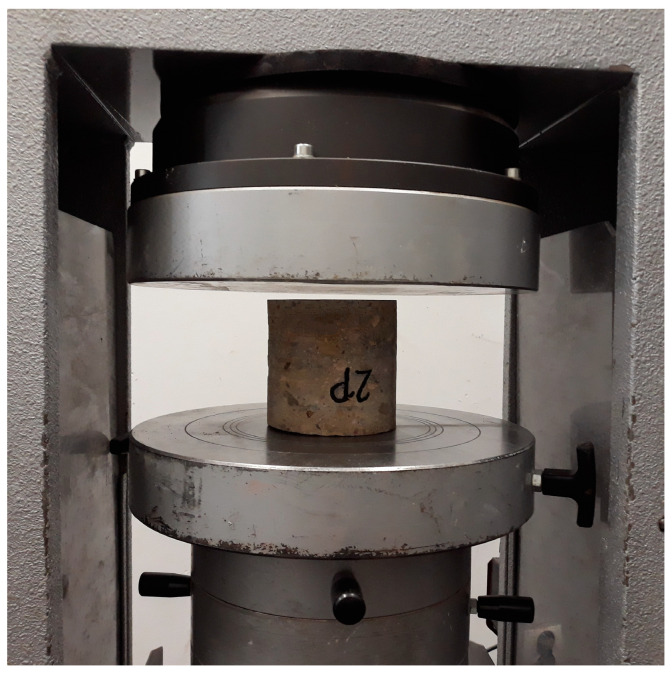
Example of a concrete sample in a strength-testing machine.

**Figure 12 materials-16-01983-f012:**
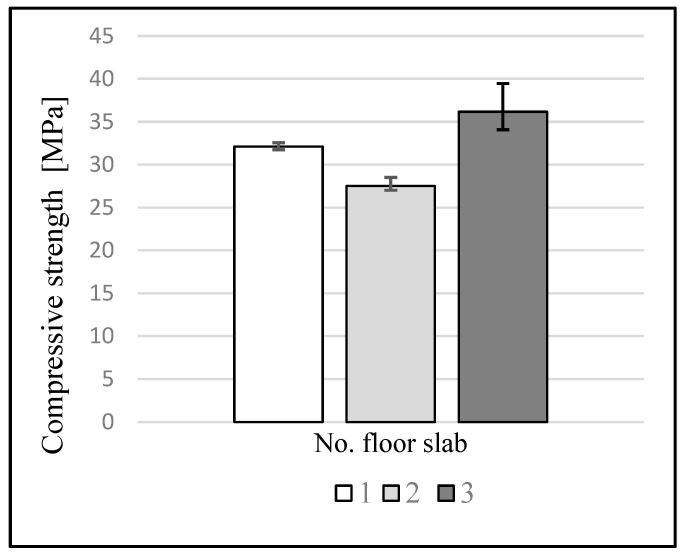
The average compressive strength of the concrete from each floor. The error bars indicate the maximum and minimum values in the floor slabs.

**Figure 13 materials-16-01983-f013:**
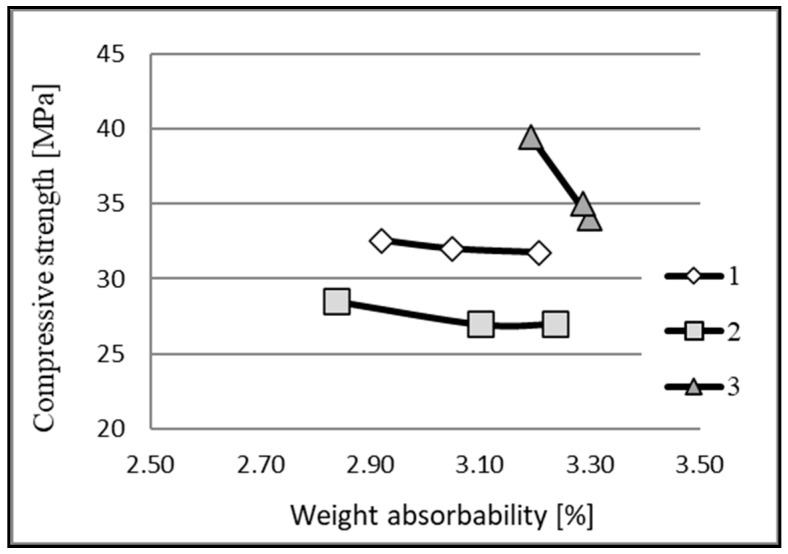
Dependence of the compressive strength on the weight absorption.

**Figure 14 materials-16-01983-f014:**
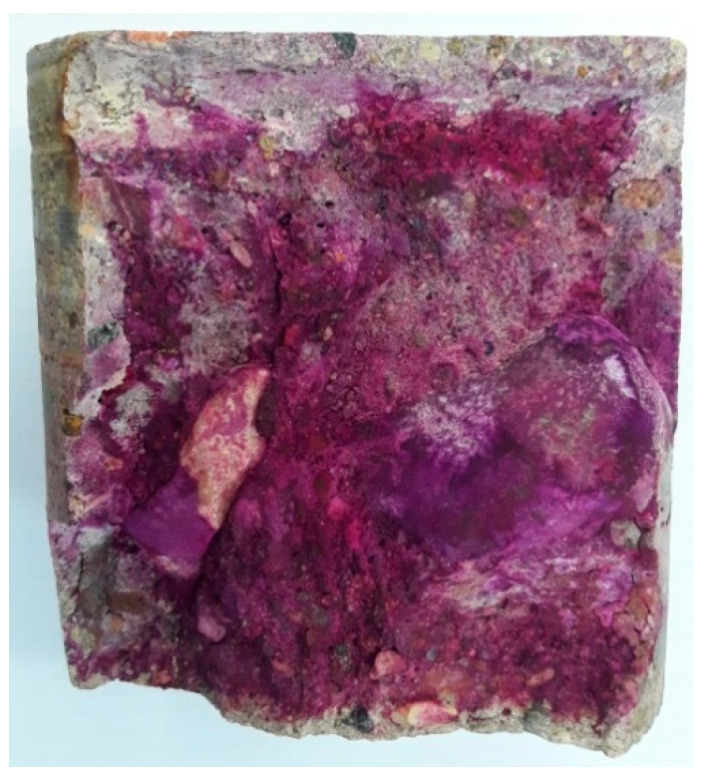
A photograph showing a concrete fracture with a phenolphthalein solution applied.

**Figure 15 materials-16-01983-f015:**
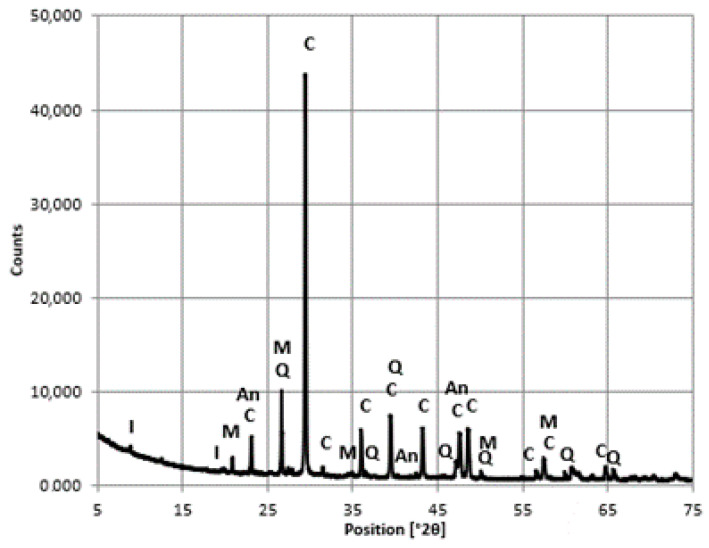
The diffractogram of the aggregate. Designation: I—illite, M—muscovite, Q—quartz, C—calcite, An—anorthoclase.

**Figure 16 materials-16-01983-f016:**
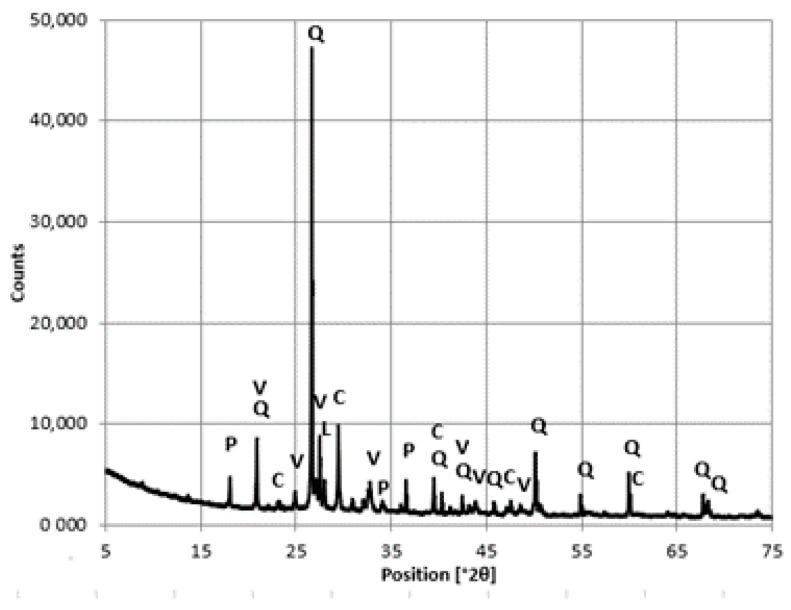
A diffractogram of the cement slurry from the concrete samples, top floor slab layer. Designation: P—portlandite, V—vaterite, A—aragonite, C—calcite, Q—quartz, L—larnite.

**Figure 17 materials-16-01983-f017:**
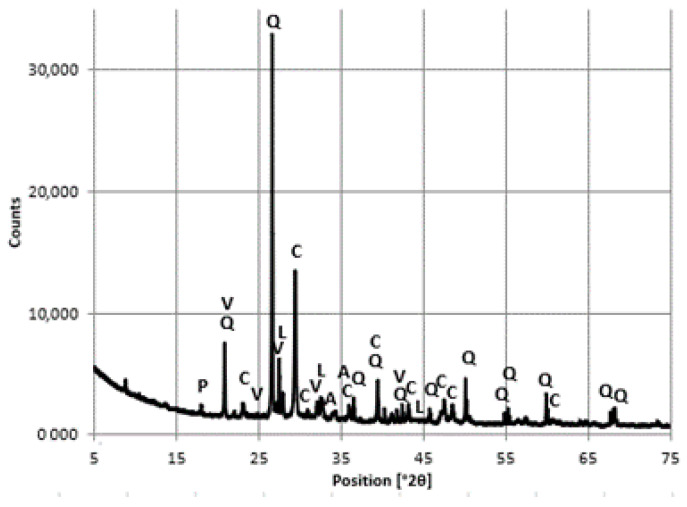
A diffractogram of the cement slurry from the concrete samples - bottom floor slab layer. Designation: P—portlandite, V—vaterite, A—aragonite, C—calcite, Q—quartz, L—larnite.

**Table 1 materials-16-01983-t001:** The physical and mechanical characteristics of concrete from floor slabs.

No. Floor Slabs	Weight Absorbability [%]	Volumetric Absorption [%]	Open Porosity [%]	Bulk Density [kg/m^3^]
1	3.06	7.43	11.49	2500
2	3.03	7.44	10.70	2528
3	3.26	8.00	12.04	2505

## Data Availability

Not applicable.

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
