# Peer review of "Analysis of the Technical Condition of a Late 19th Century Public Building in Łódź"

_materials, 2023, doi:10.3390/ma16051983_

Round 1

Reviewer 1 Report

This paper presents a study of the technical condition of a late 19th century public building in Lodz. The paper is well written and organized. There are a few minor revisions need to be conducted:

1. It is better to add scale bar to Figure 3 and Figure 9 for more explicit illustration of the structures.

2. Although the authors mentioned their testing of absorbability and porosity are aligned with the standard, the detailed steps should be included in the context or appendix, since the standard may vary from countries and it takes extra effort for readers to find. This revision is to improve the readability of the paper.

Author Response

Response to Reviewer 1 Comments

Manuscipt Title:  Analysis of the technical condition of a late 19th century public building in Łódź.

Manuscript ID: materials-2160629

Manuscript Authors: Wioletta Grzmil, Justyna Zapała-Sławeta, Jagoda Juruś

Dear Reviewer,

Thank you very much for your time and all valuable comments and suggestions about the article, as well as for re-checking the article. We have marked yellow  in the text of the article what has been corrected. We hope that our answers to the questions will be comprehensive and that the changes made to the content of the article will be correct.

Kind regards,

Wioletta Grzmil, Justyna Zapała-Sławeta and Jagoda Juruś

Reviewer 2 Report

The paper is good effort. However, there is a need to make some improvements listed below:

1. In the 'Introduction' section, clearly mention the objective of the study and the need of this study.

2. Mention the purpose of tests mentioned in line 42 and 43, one or two sentences to explain why these test are carried out.

3. In line 62 replace 'Structure' with 'structure'.

4. In section 2.1, figure 3, change the labels from A, B, C and D to Building A, Building B, Building C and Building D.

5. In Section 2.2, line 76, there is a typo 'flour', edit it to floor.

6. In Section 2.3, authors have not provided any references at all. The information in this section needs to be supported by references.

7. In line 160, correctly write CO2, 2 needs to be a sub-script.

8. In section 3.1, more discussion is needed to clearly explain the test and how they are carried out. Support use of X-Ray diffractometer with references, explain how test is carried out, why authors think this is the best test to meet study objectives. Explain Cu lamp, X'Celerator detector and why range of 5 to 75 degree was used. Overall more explanation nd more references are needed. Also for all the tests used in this study, mention the control conditions where applicable.

9. In line 204, in the unit Kg/m3, 3 should be a superscript.

10. It is suggested to add some observations how the test results are compared to show the difference or similarity of concrete used 100 years ago and these days. 

11. If authors have made assessment on remaining life of concrete, it will be a good idea to add this information.

12. If authors have taken some more pictures of concrete samples, while being tested, it will be good to add them. Like pictures of sample in testing machines etc.

13. A flowchart of research process needs to be added, showing step by step processes taken from start to completion of the research. This flowchart will go in methodology section.

Author Response

Response to Reviewer 2 Comments

Manuscipt Title:  Analysis of the technical condition of a late 19th century public building in Łódź.

Manuscript ID: materials-2160629

Manuscript Authors: Wioletta Grzmil, Justyna Zapała-Sławeta, Jagoda Juruś

Dear Reviewer,

Thank you very much for your time and all valuable comments and suggestions about the article, as well as for re-checking the article. We have marked yellow  in the text of the article what has been corrected. We hope that our answers to the questions will be comprehensive and that the changes made to the content of the article will be correct.

Kind regards,

Wioletta Grzmil, Justyna Zapała-Sławeta and Jagoda Juruś

No

Comment

Response

8

·         In section 3.1, more discussion is needed to clearly explain the test and how they are carried out. Support use of X-Ray diffractometer with references, explain how test is carried out, why authors think this is the best test to meet study objectives. Explain Cu lamp, X'Celerator detector and why range of 5 to 75 degree was used. Overall more explanation nd more references are needed. Also for all the tests used in this study, mention the control conditions where applicable.

 X-ray powder analysis is commonly used to analyze the phase composition of building materials, including the detection of reaction products formed in corrosion processes, e.g. carbonation, sulphate corrosion. Phase composition studies were performed on powder samples that were ground in an agate mortar. Then the samples were placed in a special holder in the X-ray apparatus. During the measurement, the sample rotated, i.e. it made a full rotation every 2 seconds. The measurement was made in the angle range of 5-75 2Theta, which is the standard range in which peaks from phase components present in building materials are identified (i.e. the number of reflections in this range is sufficient to confirm the occurrence of a given phase component). In this apparatus the radiation source is a Cu anode lamp. The X'Celerator is a super-fast detector - it acts as a strip of one hundred channels that can continuously count X-rays diffracted on the sample.

Panalytical software with ICDD PDF-2 database was used to analyze the diffraction patterns. The computer program compares the diffractogram obtained during the tests with the identification cards included in the catalogs and assigns the corresponding substances to the appropriate reflections. The most important information about the measurement conditions has been included in the text

10

·         It is suggested to add some observations how the test results are compared to show the difference or similarity of concrete used 100 years ago and these days. 

A description of how we compare is included in the conclusions:

            Nowadays, when designing concretes, the type of structural component and the classes of exposure of the concrete are taken into account. An analysis of the strength properties of the tested concrete and the contemporary requirements [29] indicates that the minimum class of concrete for a floor slab should be C20/25. The estimated class of the concrete from the late 19th century floor slabs, as determined for the core samples, is C25/30 for floor slabs 1 and 2, and C 30/37 for floor slab 3, respectively [30]. The specified strength classes meet the requirements for floor slabs in accordance with contemporary design recommendations.

11

If authors have made assessment on remaining life of concrete, it will be a good idea to add this information.

We did not judge.

Reviewer 3 Report

This paper presents an analysis of technical condition of a historical buildings. I think the research object of this paper is interesting and meaningful. However, the contents presented is simple and not scientific enough. 

The part of historical analysis clealy illustrates the historical evolution of the builiding, which is interesting. However, as a technical paper, the reviewer want to understand the relationships between the technical state of the building and the historical events.

The author presents a visual assessment and a test of the building. However, the results are dispersedly presented.The results and conclusion are not summerized and not well supported. On what basis does the author evaluate the technical condition of historical buildings is not clearly explained. Meanwhile, whether the author's method forms a set of system and can provide reference for later evaluation of historical buildings?

Those points should be improved in the revised manuscript. 

Author Response

Response to Reviewer 3 Comments

Manuscipt Title:  Analysis of the technical condition of a late 19th century public building in Łódź.

Manuscript ID: materials-2160629

Manuscript Authors: Wioletta Grzmil, Justyna Zapała-Sławeta, Jagoda Juruś

Dear Reviewer,

Thank you very much for your time and all valuable comments and suggestions about the article, as well as for re-checking the article. We have marked yellow  in the text of the article what has been corrected. We hope that our answers to the questions will be comprehensive and that the changes made to the content of the article will be correct.

Kind regards,

Wioletta Grzmil, Justyna Zapała-Sławeta and Jagoda Juruś
